

# Early effects of crop tree management on undergrowth plant diversity and soil physicochemical properties in a *Pinus massoniana* plantation

Qian Lyu[1], Yi Shen[1], Xianwei Li[1,2], Gang Chen[1,2], Dehui Li[3] and Chuan Fan[1,2]

[1] College of Forestry, Sichuan Agricultural University, Cheng Du, China
[2] Key Laboratory of National Forestry and Prairie Bureau on Forest Resources Conservation and Ecological Security in the Upper Reaches of Yangtze River, Sichuan Agricultural University, Cheng Du, China
[3] College of Urban and Rural Planning and Construction, Mianyang Normal University, Mian Yang, China

Corresponding author
Xianwei Li, lxw@sicau.edu.cn

## ABSTRACT

**Background**. Soil and understory vegetation are vital components of forest ecosystems. Identifying the interaction of plantation management to vegetation and soil is crucial for developing sustainable plantation ecosystem management strategies. As one of the main measures of close-to-nature management of forest plantation, few studies have paid attention to the effect of crop tree management on the soil properties and understory vegetation.

**Methods**. A 36-year-old *Pinus massoniana* plantation in Huaying city, Sichuan Province was taken as the research object to analyse the changes in undergrowth plant diversity and soil physicochemical properties under three different crop tree densities (100, 150, and 200 N/ha).

**Results**. Our results showed that the contents of available phosphorus, organic matter and hydrolysable nitrogen in the topsoil increased significantly after crop tree management, while content of available potassium decreased. The composition of shrub and herb layer was richer, and the dominant species were obviously replaced after crop tree management. The Shannon–Wiener index and Richness index of shrub layer, and the diversity of herb layer increased significantly after crop tree management. Herb layer diversity indexes and Richness index of shrub layer were closely related to soil organic matter, available phosphorus, hydrolysable nitrogen, available potassium, soil moisture and bulk density. As the main limiting factors for plant growth, nitrogen, phosphorus and potassium were closely related to plant diversity and to the distribution of the dominant species. At the initial stage of crop tree management, each treatment significantly improved the soil physicochemical properties and plant diversity of *Pinus massoniana* plantation, and the comprehensive evaluation was 200 N/ha >100 N/ha >150 N/ha >CK. Compared with other treatments, 200 N/ha had the best effect on improving the undergrowth environment of the *Pinus massoniana* plantation in the initial stage of crop tree management.

## INTRODUCTION

Forest plantations are an important component of global forest resources, and their construction has greatly increased global forest coverage and solved the conflict between the supply of and demand for wood (*Payn et al., 2015*). However, for a long time, forest plantations have grown only one tree species and have lacked scientific management techniques, resulting in ecological problems that are harmful to forest health, such as decreased plant diversity (*Paillet et al., 2010*), declining soil fertility, and frequent occurrences of diseases and insect pests (*Li et al., 2020a*). How to solve these problems and take into account the ecological and economic benefits of plantations is the focus of the current whole-plantation management strategies. According to the results of the eighth forest resources census in China, forested areas of China is 208 million ha, forest coverage rate is 21.63%, and forest plantation area is 69 million ha, which accounts for 36% of the national forest area (*Shen et al., 2019*). The forest plantation area in China accounts for approximately 33% of the world's total forest plantation area, ranking first in the world, and China is also the country with the fastest growth rate for forest plantations (*Liu et al., 2021*; *Su et al., 2021*). Chinese plantations have some problems, such as weakening carbon sequestration functions, inappropriate ecosystem structures, low productivity, and weak ecological functions and stability (*Pan et al., 2018*). Therefore, in order to improve the current situation of plantation, it is particularly important to adjust the forest plant composition and site conditions to a healthy succession process through rational forest management.

Many studies have shown that plantation management mainly changes stand density through thinning to influence light heterogeneity and stand space for undergrowth plants (*Yang et al., 2018*; *Ali et al., 2019*). The thinning intensity in boreal ecosystems is coordinated with plant diversity. Species diversity varies among different soil types, and protecting biodiversity requires a wide range of soil types (*Kershaw et al., 2015*). German close-to-nature forestry theory has achieved good results in experimental studies of various plantation forests (*Ward, 2017*). However, research on close-to-nature forest plantation management in China is still in its infancy. Crop tree management is one of the forest management methods used in close-to-nature management, and research on crop tree management in China and worldwide is mostly aimed at individual tree growth and stand growth (*Miller, 2000*; *Ray et al., 2011*; *Ward, 2017*). It is rare to study the effects of crop tree management on the function of plantation ecosystems or to reveal the mechanisms of the effects of crop tree management on plant community structure and the soil physical and chemical properties under succession. Crop tree management can be regarded as a particular kind of tending that involves thinning with a few dominant crop trees as the management centre (*Perkey, Wilkins & Smith, 1993*). Different selective cutting methods are used to cut down the disturbance trees that affect the growth of the crop trees, and it is beneficial to form gaps for the renewal of natural species and the accelerated growth of young trees under the forest canopy (*Chen et al., 2019*).

Understory plants are important components of forests because they are responsible for the majority of the vascular plant diversity of forest ecosystems (*Yilmaz, Yilmaz & Akyuz,*

*2018*). Undergrowth plant diversity can be used as an indicator of biodiversity to indicate the renewal and development of the ecosystem (*Barbier, Gosselin & Balandier, 2008*). Soil is required for the survival of forest trees and predicts changes in ecosystem multifunctionality (*Lucasborja & Delgadobaquerizo, 2019*). Soil quality conditions can interact with plant diversity; soil physicochemical properties indirectly affect plant species composition by influencing presence of soil mycorrhizal community and soil microorganisms (*Perezramos et al., 2008*). Plant diversity affects soil fertility for a long time, and an increase in soil total nitrogen may lead to the release of nitrogen limitation on certain species, thus increasing the number of plant species (*Lopezangulo et al., 2020*). Soil pH significantly affect plant species richness and change distribution range of plants due to larger species pools in high-pH soils (*Crespomendes et al., 2019*). Relatively wet and basic soils lead to high understory cover and diversity (*Kooijman et al., 2019*). Soil physicochemical properties are closely related to plant diversity. Therefore, appropriate forest management is needed to guide the ecological restoration of forest plantations to promote the development of undergrowth vegetation and improve forest productivity (*Goded et al., 2019*; *Yao et al., 2019*), which improve the ecological and physiological environment of the stand and have an impact on the diversity of plants under the forest (*Zhou et al., 2016*).

*Pinus massoniana*, one of the most important tree species in southern China, has been widely planted in this ecologically and environmentally vulnerable area because of its high adaptability to drought and barren soils and its capacity to retain water and nutrients (*Ma et al., 2014*). *Pinus massoniana* plantations cover approximately 10 million ha and account for approximately 20% of the planted forests in southern China (*Yu et al., 2019*). An appropriate forest quality improvement plan is an urgent scientific and technological need for securing the ecological restoration achievements in low mountainous areas of eastern Sichuan. Crop tree management can be used to scientifically manage and transform *Pinus massoniana* plantations; it can not only meet human demand for wood but also improve plant community structure, increase the diversity of plants in the forest understory and simultaneously allow the various benefits of the plantation to develop fully. In this study, taking a *Pinus massoniana* plantation as the research object, different crop tree density treatments were used to explore the effects of crop tree management on plant diversity and soil physicochemical properties. The goals of this study are to identify the optimal crop tree density for the natural restoration of local plantations and to provide a reference for the selection and study of management practices for *Pinus massoniana* plantations.

## MATERIALS & METHODS

### Study site

The study area is located at Tianchi Forest Farm, Hua Ying city, Sichuan Province, in the low mountainous area of eastern Sichuan. The area is subtropical and in the middle of a humid monsoon climate region (106°47′30″E, 30°20′29″N, Fig. 1). The climate is mild, with uneven rainfall and large temperature differences. The multiyear average temperature in Huaying is 17.2 °C. The rainfall is abundant; the highest annual average rainfall is 1441.7 mm, the lowest annual average rainfall is 854.9 mm, and the multiyear average rainfall is

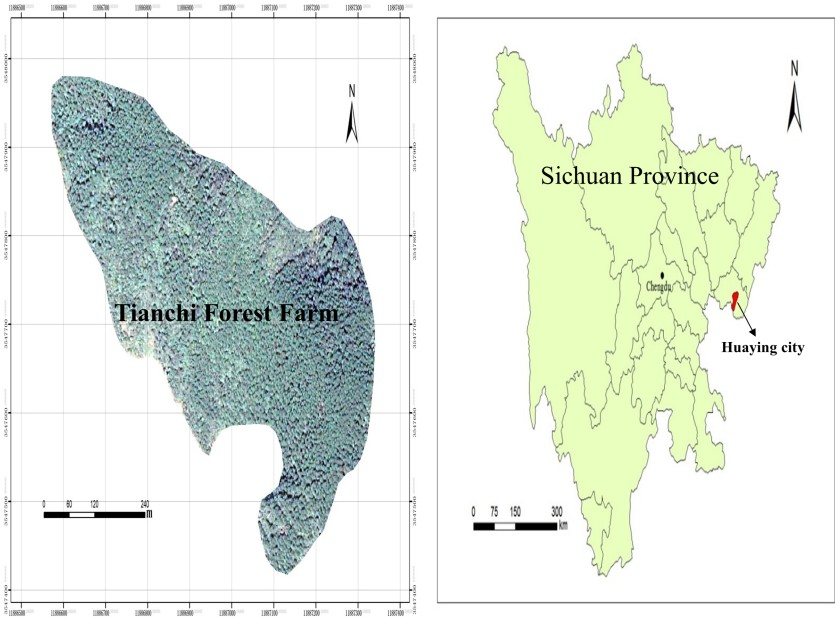

**Figure 1  Location of the study area.**

1087.84 mm. Soil is typical yellow soil with poor tillage and low fertility. *Pinus massoniana* plantation was established in 1982. Although necessary management measures have been performed in the stands, the overall management level has been low. In the crop tree plot of *Pinus massoniana* plantation, due to the ground is covered with a large number of *Pinus massoniana* needles, plant diversity under the forest canopy is poor. Shrubs are mainly *Litsea pungens* Hemsl, *Mallotus barbatus* (Wall.) and *Quercus serrata* Thunb, and the herbaceous layer is mainly covered with ferns.

## Experimental design and sampling

In January 2015, site conditions were basically same throughout the study area according to the principle of typical sampling. The 33-year-old near-mature forest stand that had the same forest age and management history was managed as crop trees, but few management measures were taken before this study. Three crop tree densities (100, 150, and 200 N/ha, marked as A, B, and C, respectively) were established. A designated sample plot without crop tree management was also established as the control plot (CK). Three 30 m × 30 m sample squares were set up in *Pinus massoniana* plantation with different management modes. There were 12 sample plots in total. Trees with vigorous growth, straight trunks, well-developed crowns, and no damage, diseases or insect pests that were located in the main forest layer were selected as the crop trees and were permanently labelled with red flagging tapes. At the same time, any competitive trees that affected the growth of the crop tree were identified to prevent them from touching the crop tree. An overview of a crop tree sample plot is shown in Table 1. In the crop tree management sample plot, the general trees (non-crop trees) were marked with flagging tapes and the boundary trees were

**Table 1 Growth of *Pinus massoniana* before and after crop tree management.** A (crop tree densities of 100 N/ha), B (crop tree densities of 150 N/ha), C (crop tree densities of 200 N/ha).

| Treatment | Before crop tree management | | | After crop tree management | | |
|---|---|---|---|---|---|---|
| | DBH (cm) | Tree height (m) | Density of crop trees (N/ha) | DBH (cm) | Tree height (m) | Density of crop trees (N/ha) |
| CK | 19.84 ± 0.04 | 10.05 ± 0.02 | – | – | – | – |
| A | 19.00 ± 0.04 | 10.56 ± 0.06 | – | 19.54 ± 0.37 | 10.50 ± 0.06 | 100 |
| B | 19.80 ± 0.30 | 12.25 ± 0.32 | – | 20.01 ± 0.10 | 12.02 ± 0.07 | 150 |
| C | 19.70 ± 0.20 | 11.15 ± 0.15 | – | 19.91 ± 0.09 | 11.03 ± 0.02 | 200 |

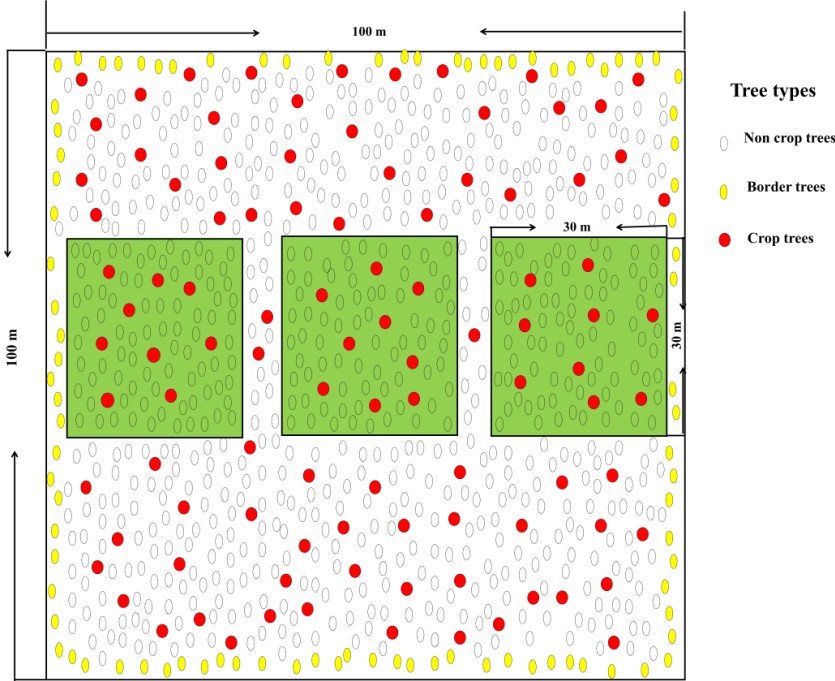

**Figure 2 Samplings plots with a crop tree density of 150 N/ha.**

marked with yellow flagging tapes. Protective fences were established around the sample plots (Fig. 2).

## Undergrowth species survey

Undergrowth plants in the sample plots were surveyed in October 2018. Five 5 m × 5 m quadrats were established shrub survey in the four corners of each treatment and in the centre of each of the sample plots in *Pinus massoniana* plantation. Then, ten 1 m × 1 m quadrats were randomly established for the herbaceous survey in the sample plots. The species name, number of plants, height, coverage, crown width and other data for each plant in the shrub and herb sample quadrats were recorded. According to the data obtained from the sample plot survey, the importance values of the plants in the sample

plots were calculated, and then the diversity index was calculated. The species diversity indicators included the species importance value (IV), richness index (R), species number (S), Simpson index (D), Shannon–Wiener index (H), and Pielou index (J) (*Wang et al., 2019*). The formulas are as follows:

$$\text{Important value}(\%): IV = (\text{Relative density} + \text{Relative frequency} +$$
$$\text{Relative coverage})/3 \tag{1}$$

$$R = S \tag{2}$$

$$\text{Simpson index}: D = 1 - \sum_{i=1}^{s} P_i^2 \tag{3}$$

$$\text{Shannon} - \text{Wiener index}: H = - \sum_{i=1}^{s} P_i \ln P_i \tag{4}$$

$$\text{Pielou index}: J = H/\ln S \tag{5}$$

Note: "S", total number of species in the sample plot; "$P_i$", the proportion of the number of individuals of one species to the sum of the number of individuals of all species in this layer (*Alatalo, 1981*).

Niche breadth was calculated based on the Shannon–Wiener niche breadth index (*Yadav et al., 2015*):

$$B_i = - \sum_{j=1}^{r} P_{ij} \ln P_{ij} \tag{6}$$

Note: "$B_i$", the niche breadth of species "i"; $P_{ij} = n_{ij}/N_i$, "$n_{ij}$", the importance value of species "I" for resource "j"; "$N_i$", the sum of the importance values of species "i" for all resources. The importance value of "$P_{ij}$" represents species "i" for resource "j" and accounts for the proportion of the importance value of this species for all resources; "r", the total number of resource units; "j", the different transformation modes (*Tonkin, Arimoro & Haase, 2016*).

## Soil investigation

A soil drill (five cm in diameter) was used in each crop tree management stand to randomly arrange five sampling points in an "S" shape to collect soil samples. Soil samples at depths of 0~10 cm and 10~20 cm were collected separately at each sampling point and mixed evenly. Approximately 1.5 kg of each sample was selected according to the quartet method, packed into plastic bags and taken back to the laboratory within 24 h. Fine roots, fine gravel and litter were separated from the soil manually. Then, the soil samples were air-dried and

passed through two mm and 0.149 mm sieves. A subsample was dried at 105 °C for 48 h to constant weight to measure the soil gravimetric moisture content and soil bulk density (*Luo et al., 2018*). The organic matter content of the soil was determined by hydration with the potassium dichromate oxidation-colorimetric method (*Yang et al., 2020*). The soil pH was measured in a 1:2.5 water and soil mixture whisked together for 10 min with a glass rod, followed by standing for 1 h and then being measured with an electronic pH metre (*Li et al., 2020a*). Soil hydrolysable nitrogen was measured by the alkali-hydrolysed diffusion absorption method (*Chen et al., 2010*). Available potassium was measured by the ammonium acetate extraction-flame photometric detection method (*Rich, 1958*). Soil available phosphorus was measured by extracting subsamples with 0.03 M NH4F – 0.025 M HCl (*Mariotte et al., 2020*).

## Statistical analyses

In this study, Excel 2010 was used for data processing, origin Pro 8.0 was used to create the figures, and SPSS 20.0 was used for statistical analysis. Single factor analysis of variance (one-way ANOVA) and multiple comparisons (LSD) were used to test the soil index and plant diversity index in the *Pinus massoniana* forest under the different treatments. Two-way ANOVA was used to test whether there was a significant interaction among the different crop tree treatments, different soil layers and soil physicochemical properties. Pearson correlation analysis was carried out between the plant diversity index and the soil physicochemical properties. The principal component variables were extracted from the comprehensive indexes of soil physicochemical properties and plant diversity by principal component analysis (PCA), a comprehensive evaluation value was calculated through the comprehensive evaluation model. Pearson correlation analysis and principal component analysis (PCA) was performed using the OmicShare tools, a free online platform for data analysis (http://www.omicshare.com/tools). An RDA of the environmental factors and dominant species was performed with CANOCO 5.0.

$$F = F1 \times \frac{\lambda 1}{\lambda 1 + \lambda 2 + \lambda 3} + F2 \times \frac{\lambda 2}{\lambda 1 + \lambda 2 + \lambda 3} + F3 \times \frac{\lambda 3}{\lambda 1 + \lambda 2 + \lambda 3}$$

In the formula, F1, F2 and F3 are the main components, $\lambda 1$, $\lambda 2$ and $\lambda 3$ are the characteristic root values, and F is the comprehensive evaluation value.

# RESULTS

## Plant composition and undergrowth plant diversity in the initial stage of crop tree management

There were more plant species in the shrub layer under crop tree management than under the control treatment (Table 2). In terms of shrub layer, C treatment has the largest number of species among all treatments. The dominant species under the three crop tree treatments changed compared with those under the control treatment. The dominant species in the control treatment were *Cinnamomum camphora* (Linn.), *Debregeasia orientalis*, *Litsea pungens*, *Mallotus barbatus*, *Quercus serrata* and *Urena lobata* (Linn.). The dominant species in the A treatment were *Cinnamomum camphora*, *Eurya brevistyla* Kobuski, *Litsea*

**Table 2 The importance value (IV/%) and niche breadth (Bi) of the shrub layer in a *P. massoniana* plantation in the early stage of crop tree management.** A (crop tree densities of 100 N/ha), B (crop tree densities of 150 N/ha), C (crop tree densities of 200 N/ha).

| Serial number | Species | Treatment | | | | | | | |
| --- | --- | --- | --- | --- | --- | --- | --- | --- | --- |
| | | CK | | A | | B | | C | |
| | | IV | Bi | IV | Bi | IV | Bi | IV | Bi |
| 1 | *Myrsine africana* | – | – | 3.13 | 1.4 | 5.42 | 1.44 | 3.45 | 1.43 |
| 2 | *Cinnamomum camphora* | 18.25 | 1.38 | 8.51 | 1.36 | 4.98 | 1.09 | 4.42 | 1.03 |
| 3 | *Rubus pirifolius* | 3.92 | 1.43 | – | – | 4 | 1.43 | 6.06 | 1.45 |
| 4 | *Mallotus barbatus* | 16.07 | 1.36 | 8.09 | 1 | 23.03 | 1.46 | 22.13 | 1.46 |
| 5 | *Aralia chinensis* | – | – | – | – | 1.04 | 1.16 | 1.78 | 3.72 |
| 6 | *Eurya brevistyla* | 4.17 | 1.29 | 8.59 | 1.46 | 5.11 | 1.38 | 3.01 | 1.12 |
| 7 | *Smilax china* | – | – | – | – | 2.9 | 1.29 | 2.24 | 1.45 |
| 8 | *Litsea pungens* | 11.96 | 1.28 | 17.52 | 1.44 | 16.21 | 1.41 | 14.64 | 1.37 |
| 9 | *Callicarpa giraldii* | – | – | 3.65 | 0.71 | – | – | 0.91 | 1.29 |
| 10 | *Melastoma malabathricum* | 5.08 | 1.41 | 3.35 | 1.23 | 4.31 | 1.35 | 6.07 | 1.46 |
| 11 | *Ardisia japonica* | – | – | 4.78 | 1.38 | 2.34 | 1.38 | 2.41 | 1.39 |
| 12 | *Quercus serrata* | 10.41 | 1.47 | 5.57 | 1.28 | 6.3 | 1.34 | 5.73 | 1.3 |
| 13 | *Serissa japonica* | 4.1 | 0.93 | – | – | – | – | 1.54 | 1.42 |
| 14 | *Rubus buergeri* | – | – | – | – | – | – | 1.89 | 0 |
| 15 | *Smilax discotis* | – | – | – | – | 2.94 | 1.25 | 2.05 | 1.46 |
| 16 | *Gardenia jasminoides* | 2.98 | 1.4 | 3.23 | 1.43 | 1.84 | 1.17 | 3.48 | 1.45 |
| 17 | *Ficus pandurata* | 3.95 | 1.34 | 5.78 | 1.46 | 4.58 | 1.4 | 3.53 | 1.28 |
| 18 | *Pithecellobium lucidum* | – | – | – | – | – | – | 0.97 | 0 |
| 19 | *Urena lobata* | 7.82 | 1.27 | 3.61 | 1.41 | 1.34 | 0.91 | 0.84 | 0.69 |
| 20 | *Smilax arisanensis* | – | – | – | – | 2.13 | 1.31 | 1.71 | 1.44 |
| 21 | *Callicarpa bodinieri* | – | – | – | – | – | – | 0.82 | 0 |
| 22 | *Rhododendron simsii* | – | – | 3.1 | 1.17 | – | – | 1.85 | 1.47 |
| 23 | *Rubus corchorifolius* | – | – | 4.12 | 1.3 | 1.59 | 1.32 | 1.66 | 1.34 |
| 24 | *Lonicera japonica* | – | – | – | – | – | – | 1.08 | 0 |
| 25 | *Mallotus philippensis* | – | – | – | – | – | – | 0.78 | 0 |
| 26 | *Ardisia crispa* | – | – | – | – | – | – | 1.62 | 0 |
| 27 | *Pericampylus glaucus* | – | – | 4.88 | 1.47 | 4.13 | 1.47 | 3.34 | 1.41 |
| 28 | *Ampelopsis delavayana* | – | – | 3.07 | 0.87 | 1.03 | 1.39 | – | – |
| 29 | *Millettia congestiflora* | – | – | 4 | 1.37 | 2.88 | 1.47 | – | – |
| 30 | *Debregeasia orientalis* | 9.01 | 0 | – | – | – | – | – | – |
| 31 | *Rhus chinensis* | 2.28 | 0 | – | – | – | – | – | – |
| 32 | *Symplocos lancifolia* | – | – | 3.41 | 0 | – | – | – | – |
| 33 | *Broussonetia kaempferi* | – | – | 4.01 | 0 | – | – | – | – |
| 34 | *Lespedeza bicolor* | – | – | – | – | 1.88 | 0 | – | – |
| 35 | *Dalbergia hupeana* | – | – | 6.26 | 0 | | | – | – |
| 36 | *Ficus henryi* | – | – | 3.17 | 0 | – | – | – | – |

*pungens*, *Mallotus barbatus* and *Quercus serrata*. The dominant species in the B treatment were *Litsea pungens*, *Mallotus barbatus*, *Myrsine africana* (Linn.) and *Quercus serrata*. The dominant species in the C treatment were *Litsea pungens*, *Mallotus barbatus*, *Melastoma malabathricum* Linnaeus and *Rubus pirifolius* (Table 2).

Crop tree management promoted the emergence of different herbaceous plant species (Table 3). The proportion of pteridophytes in the control was relatively high, but with the increase in the intensity of the different treatments, the proportion of pteridophytas gradually decreased, which promoted the emergence of more different herbaceous species. The dominant species under the three crop tree treatments changed compared with those under the control treatment. The dominant species in the control treatment were *Dryopteris fuscipes* C. Chr, *Miscanthus sinensis*, *Iris tectorum* Anderss, and *Setaria plicata* (Lam.). The dominant species in the A treatment were *Dicranopteris dichotoma* (Thunb.), *Iris tectorum* and *Setaria plicata*. The dominant species in the B treatment were *Dryopteris fuscipes*, *Microlepia hancei* Prantl, *Miscanthus sinensis* and *Setaria plicat* a. The dominant species in the C treatment were *Microlepia hancei*, *Miscanthus sinensis*, *Dicranopteris dichotoma* and *Stenoloma chusanum* Ching (Table 3).

In the shrub layer, crop tree management on *Pinus massoniana* plantation had a significant effect on the Pielou index, Richness index, and Shannon–Wiener index ($p < 0.05$), but had no significant effect on the Simpson index (Fig. 3). Compared to the control, the Richness index of crop tree management increasing by 61.54%, 61.54% and 107.69% for treatments A, B and C, respectively, and the Shannon–Wiener index of crop tree management increasing by 21.17%, 21.20% and 23.27%, for treatments A, B and C, respectively. The Shannon–Wiener index and Richness index of the shrub layer were the highest in the C treatment. In the herb layer, the Pielou index, Richness index, Shannon–Wiener index and Simpson index of the herb layer were significantly higher than those of the control ($p < 0.05$) (Fig. 3). The Pielou index and Simpson index of herb layer were C >A >B >CK. The Richness index of crop tree management was 1.17 to 1.25 times higher than control. Compared to the control, The Shannon–Wiener index of crop tree management, increasing by 15.90%, 10.63% and 15.82%, for treatments A, B and C, respectively.

## Effects of crop tree management on soil physicochemical properties

Table 4 shows that the effect of crop tree management on the physicochemical properties of topsoil in *Pinus massoniana* plantation is significantly stronger than that on deep soil properties. In the 0∼10 cm soil layer, compared with those under CK, the contents of soil organic matter and available phosphorus increased significantly ($p < 0.05$), while soil available potassium and soil bulk density decreased significantly ($p < 0.05$). Under treatments A and C, the content of soil hydrolysable nitrogen was significantly higher than under CK ($p < 0.05$); however, that under treatment B was significantly lower than that under CK ($p < 0.05$). In the 10∼20 cm soil layer, the treatments had no significant effect on soil pH, organic matter, soil moisture or soil bulk density, but the contents of available phosphorus and hydrolysable nitrogen in soil increased significantly compared with those under CK ($p < 0.05$).

**Table 3  The importance value (IV/%) and niche breadth (Bi) of the herb layer in the early stage of crop tree management in a *Pinus massoni-ana* plantation.** A (crop tree densities of 100 N/ha), B (crop tree densities of 150 N/ha), C (crop tree densities of 200 N/ha).

| Serial number | Species | Treatment | | | | | | | |
|---|---|---|---|---|---|---|---|---|---|
| | | CK | | A | | B | | C | |
| | | IV | Bi | IV | Bi | IV | Bi | IV | Bi |
| 37 | *Lophatherum gracile* | 2.85 | 1.17 | 6.24 | 1.47 | 2.28 | 1.05 | 6.47 | 1.47 |
| 38 | *Woodwardia japonica* | 4.83 | 1.30 | – | – | 9.42 | 1.47 | 9.43 | 1.47 |
| 39 | *Dendranthema indicum* | 3.78 | 0.88 | 1.31 | 1.40 | – | – | – | – |
| 40 | *Dryopteris fuscipes* | 11.79 | 1.39 | 8.90 | 1.27 | 22.06 | 1.41 | 3.35 | 0.76 |
| 41 | *Parathelypteris glanduligera* | 6.62 | 0.00 | – | – | – | – | – | – |
| 42 | *Erigeron annuus* | 1.56 | 1.44 | – | – | – | – | 3.82 | 0.97 |
| 43 | *Setaria plicata* | 28.53 | 1.42 | 13.81 | 1.35 | 10.52 | 1.22 | 7.56 | 1.04 |
| 44 | *Oxalis corniculata* | 6.62 | 0.94 | – | – | 2.57 | 1.42 | – | – |
| 45 | *Sanguisorba officinalis* | – | – | 1.31 | 0.00 | – | – | – | – |
| 46 | *Cyrtomium fortunei* | – | – | 2.61 | 0.00 | – | – | – | – |
| 47 | *Lygodium japonicum* | – | – | 2.85 | 1.17 | 1.71 | 1.47 | – | – |
| 48 | *Capillipedium parviflorum* | – | – | – | – | 2.50 | 0.00 | – | – |
| 49 | *Dicranopteris dichotoma* | – | – | 11.19 | 1.45 | 4.36 | 1.17 | 11.15 | 1.46 |
| 50 | *Blumea megacephala* | – | – | – | – | – | – | 7.09 | 0.00 |
| 51 | *Alpinia japonica* | – | – | – | – | – | – | 1.64 | 0.00 |
| 52 | *Oplismenus compositus* | – | – | – | – | – | – | 3.57 | 0.00 |
| 53 | *Liriope spicata* | – | – | 2.91 | 1.46 | – | – | 4.13 | 1.25 |
| 54 | *Stenoloma chusanum* | – | – | 6.00 | 1.46 | – | – | 12.86 | 1.04 |
| 55 | *Dioscorea opposita* | – | – | – | – | 1.75 | 0.00 | – | – |
| 56 | *Microlepia hancei* | – | – | 6.58 | 1.31 | 12.70 | 1.46 | 12.17 | 1.47 |
| 57 | *Arthraxon hispidus* | 3.98 | 1.32 | 9.91 | 1.34 | 2.28 | 1.03 | 2.33 | 1.04 |
| 58 | *Pteridium aquilinum* | 4.20 | 1.28 | 7.12 | 1.47 | 2.28 | 0.96 | 7.56 | 1.47 |
| 59 | *Miscanthus sinensis* | 13.02 | 1.44 | 6.88 | 1.15 | 14.20 | 1.46 | 10.44 | 1.36 |
| 60 | *Iris tectorum* | 12.23 | 1.47 | 11.59 | 1.47 | 7.81 | 1.38 | – | – |

Except for the index of pH, crop tree management had significant effect on other soil properties (Table 5, $p < 0.05$). Only soil available potassium, soil moisture and soil bulk density were significantly affected by different soil layers (Table 5, $p < 0.05$). The interactions of crop tree management and the different soil layers had significant effects on soil hydrolysable nitrogen and soil bulk density (Table 5, $p < 0.05$).

### Relationships between plant characteristics and soil factors

As seen in Fig. 4, the correlation coefficient between herb layer plant diversity index and soil properties was higher than that between shrub layer plant diversity and soil properties. There was no correlation between D1 and soil physical or chemical properties. H1 was significantly positively correlated with SP ($P < 0.01$), while significantly negatively correlated with AK ($p < 0.05$). J1 was negatively correlated with SOM and SM ($p < 0.05$), but positively correlated with AK and BD ($p < 0.05$). R1 was significantly positively correlated with SOM, AP, AN and SM, but significantly negatively correlated with BD and AK ($p < 0.05$). D2 had highly significant positive correlation with SM and SP ($P < 0.01$),

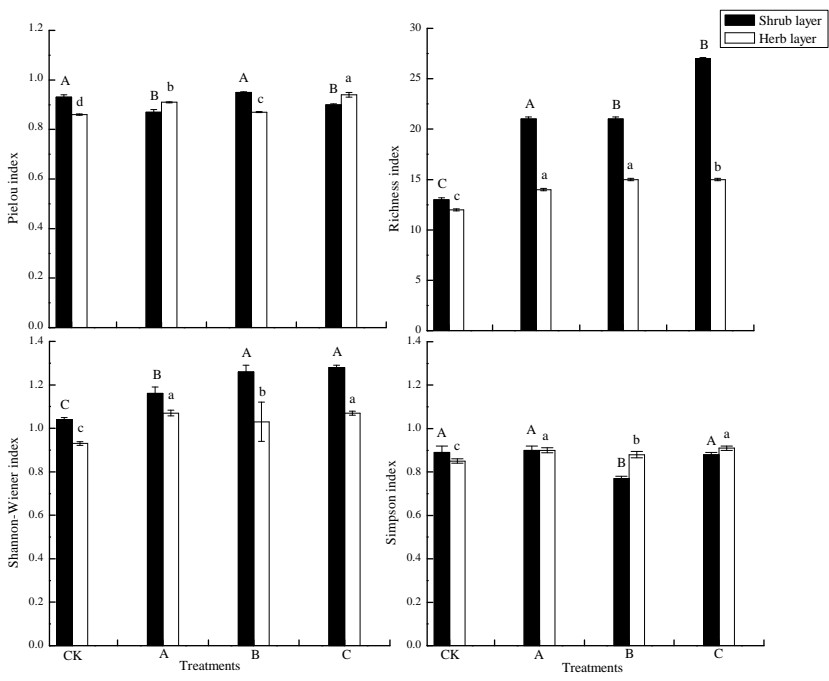

**Figure 3 Plant diversity under different treatments.** (A) Crop tree densities of 100 N/ha, (B) crop tree densities of 150 N/ha, (C) crop tree densities of 200 N/ha. Different capital letters indicate significant differences among treatments in the shrub layer at $p < 0.05$. Different lowercase letters indicate significant differences among treatments in the herb layer at $p < 0.05$.

but significantly negatively correlation with AK and BD ($P < 0.01$). There was a positive correlation between D2 and SM ($p < 0.05$). H2 was highly positively correlated with SOM and SP ($p < 0.01$), while highly negatively correlated with SK, SM or BD ($p < 0.01$). J2 was highly significantly positively correlated with SOM and AN ($p < 0.01$), while highly negatively correlated with AK and BD ($p < 0.01$). S2 was highly significantly positively correlated with SOM ($p < 0.01$), while highly negatively correlated with AK and BD ($p < 0.01$).

Figure 5A presented the results of the total explanation of the effect of soil physical and chemical properties on shrub layer species under 0~10 cm soil layer was 74.14%, which could better reflect the relationship between shrub layer species and environmental factors, in which AK had a very significant effect on shrub layer species ($p < 0.01$). Figure 5B showed the RDA analysis of shrub layer under 10~20 cm soil layer. At this level, RDA1 explained 54.42% of all information. RDA2 explained 21.84% of all information. The total degree of explanation of the two was 76.26%. It reflected the actual situation of shrub layer species and environmental factors, in which AK had a very significant impact on shrub layer species ($p < 0.01$), and AN had a significant impact on shrub layer species ($p < 0.05$). Figure 5C showed that the total explanation of the soil physical and chemical properties of the 0~10 cm soil layer to the herb layer species was up to 70.05%. AK, AN and AP all had significant effects on the herb layer species ($p < 0.01$). Figure 5D showed the RDA analysis

Lyu et al. (2021), *PeerJ*, DOI 10.7717/peerj.11852

**Table 4** **Soil physicochemical properties under different soil layers in the early stage of crop tree management.** A (crop tree densities of 100 N/ha), B (crop tree densities of 150 N/ha), C (crop tree densities of 200 N/ha). Different small letters indicate significant differences among treatments in $p < 0.05$.

| Soil layers | Treatment | pH | SOM (mg/kg) | AP (mg/kg) | AN (mg/kg) | AK (mg/kg) | SM (%) | BD (g/cm³) |
|---|---|---|---|---|---|---|---|---|
| 0–10 cm | CK | 3.67 ± 0.04a | 18.86 ± 9.55c | 1.32 ± 0.29b | 271.21 ± 52.36bc | 80.58 ± 1.48a | 23.80 ± 0.74b | 1.40 ± 0.01a |
| | A | 3.43 ± 0.12a | 34.88 ± 1.88ab | 4.65 ± 0.93a | 314.58 ± 23.07b | 64.56 ± 0.96d | 27.19 ± 1.33a | 1.38 ± 0.02b |
| | B | 3.46 ± 0.15a | 22.91 ± 5.09bc | 5.95 ± 1.28a | 208.00 ± 81.27c | 72.32 ± 1.89b | 23.73 ± 0.77b | 1.36 ± 0.03b |
| | C | 3.50 ± 0.14a | 39.03 ± 6.66c | 5.37 ± 1.59a | 481.60 ± 0.21a | 68.36 ± 2.57c | 26.67 ± 0.25a | 1.33 ± 0.01b |
| 10–20 cm | CK | 3.67 ± 0.04a | 24.69 ± 1.28a | 2.75 ± 3.10b | 247.79 ± 41.38c | 76.74 ± 3.02a | 24.41 ± 3.27a | 1.41 ± 0.03a |
| | A | 3.43 ± 0.12a | 19.40 ± 3.96a | 4.95 ± 1.43ab | 321.03 ± 34.87b | 59.11 ± 0.63b | 24.11 ± 0.60a | 1.40 ± 0.02a |
| | B | 3.46 ± 0.15a | 29.32 ± 11.97a | 6.70 ± 0.42a | 351.37 ± 36.24ab | 63.08 ± 5.56b | 23.05 ± 0.67a | 1.41 ± 0.02a |
| | C | 3.47 ± 0.14a | 32.46 ± 12.80a | 4.31 ± 0.89ab | 401.24 ± 0.33a | 61.84 ± 1.00b | 23.97 ± 0.19a | 1.41 ± 0.01a |

**Table 5  Results of two-way ANOVA for the effects of crop tree management (CTM), soil layers (SL), and their interactions (CTM SL) on soil physicochemical properties. (DF: Degrees of freedom).** A (crop tree densities of 100 N/ha), B (crop tree densities of 150 N/ha), C (crop tree densities of 200 N/ha).

| Index | | CTM | SL | CTM × SL |
|---|---|---|---|---|
| | F | 2.506 | 0.695 | 0.108 |
| pH | P | 0.096 | 0.417 | 0.954 |
| | DF | 3 | 1 | 3 |
| | F | 3.356 | 0.589 | 2.726 |
| SOM | P | 0.045 | 0.454 | 0.079 |
| | DF | 3 | 1 | 3 |
| | F | 8.682 | 0.341 | 0.748 |
| AP | P | 0.001 | 0.568 | 0.539 |
| | DF | 3 | 1 | 3 |
| | F | 22.604 | 0.450 | 7.669 |
| AN | P | <0.001 | 0.512 | 0.002 |
| | DF | 3 | 1 | 3 |
| | F | 46.701 | 34.428 | 1.130 |
| AK | P | <0.001 | <0.001 | 0.367 |
| | DF | 3 | 1 | 3 |
| | F | 3.675 | 7.070 | 2.512 |
| SM | P | 0.035 | 0.017 | 0.096 |
| | DF | 3 | 1 | 3 |
| | F | 7.879 | 52.364 | 5.576 |
| BD | P | 0.002 | <0.001 | 0.008 |
| | DF | 3 | 1 | 3 |

of herb layer under 10∼20 cm. At this level, RDA1 explained 52.44% of all information. RDA2 explained 21.52% of all information. The total degree of explanation of the two was 73.96%. It reflected the actual situation of herb layer species and environmental factors, in which AN had a very significant impact on shrub layer species ($p < 0.01$).

A total of 15 indexes of plant diversity and surface soil physicochemical properties under crop tree management on *Pinus massoniana* plantation were analysed by principal component analysis. As shown in Table 6, three principal component variables were extracted, and the cumulative contribution rate reached 89.67% (Table 6). Therefore, the comprehensive index of the three extracted principal components can be used to summarize and represent the 15 individual indicators. According to the comprehensive evaluation of the principal components, the close-to-nature management effects of the three crop tree treatment plots was significantly higher than that in the control plots (Fig. 6), and of all the treatments, treatment C had the most significant effect on the ecosystem service functions of *Pinus massoniana* plantation (Table 7).

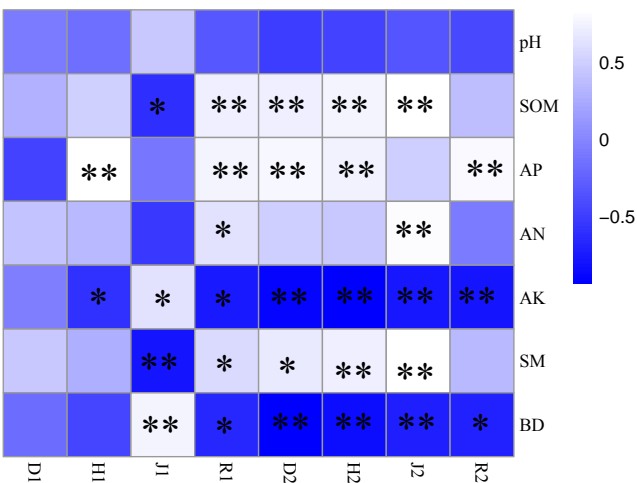

**Figure 4** **Correlation coefficient of plant diversity and soil physical and chemical properties under different crop tree management.** R1, the richness index of the shrub layer; R2, the richness index of the herb layer; H1, the Shannon–Wiener index of the shrub layer; H2, the Shannon–Wiener index of the herb layer; D1, the simpson index of the shrub layer; D2, the Simpson index of the herb layer; J1, the Pielou index of the shrub layer; J2, the Pielou index of the herb layer (* $P < 0.05$, ** $P < 0.01$).

## DISCUSSION

### Effects of the early stage of crop tree management on undergrowth plant diversity and soil physicochemical properties

The composition and structure of the forest community have strong influences on the forest ecosystem (*Ding et al., 2017*). Increasing plant diversity is increasingly regarded as the most effective means of improving the ecosystem functions of forest plantations, and plants with various life forms can better maintain the stability of forest ecosystems. In the context of climate warming and the increasing demand for wood products, there must be a better understanding of the proper management of biodiversity and ecosystem functions (*Schulze et al., 2016*). The plant composition and diversity under the three different crop tree densities were more abundant than that in the control treatment, which may be due to the fact that the thinning of disturbed trees can improve the light conditions in the stand by reducing the stand density and canopy density, increasing the vegetative area and growth space of plants, promoting the growth and development of shrubs and herbs under the forest, and making the composition and site conditions of forest vegetation develop in a healthy succession direction (*Romeo et al., 2020*; *Sanaei et al., 2021*). And most studies have found that the increase in plant diversity were related to light and hydrothermal conditions in forests, thereby directly or indirectly affected the characteristics of aboveground and underground forests (*Deng et al., 2020*; *Aun et al., 2021*; *Gong et al., 2021*). The crop tree management was also beneficial to optimize the species composition of the stand. The C treatment had the most plant species, and the plant pattern showed a transition from light-requiring plants to light-neutral plants and then to shade plants. The niche breadth of the light-loving plants such as *Cinnamomum camphora* (Linn.), *Quercus serrata* Thunb

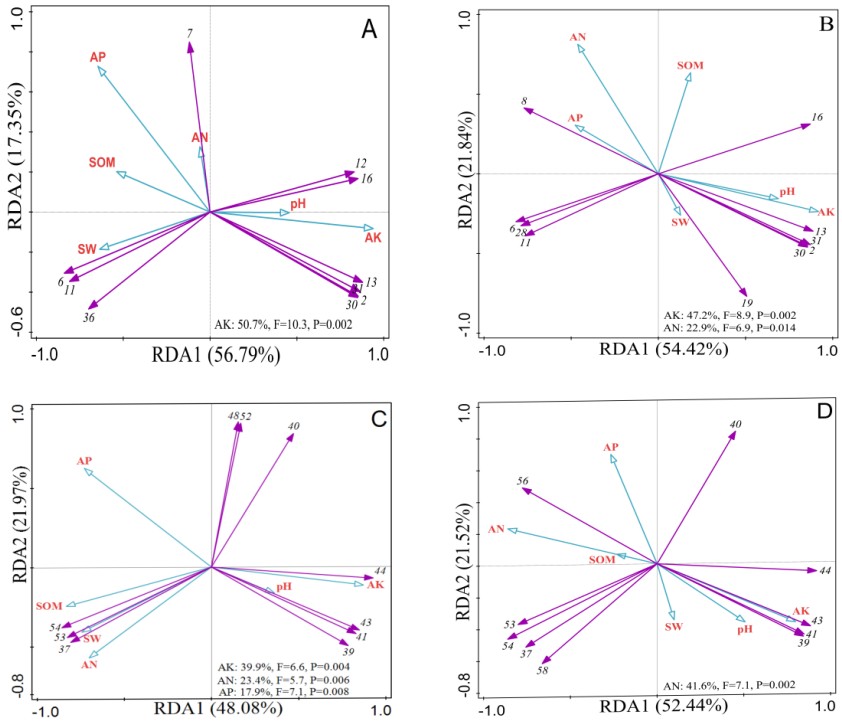

**Figure 5 RDA analysis plot showing the relationships between soil physicochemical properties and shrub and herb composition at soil depth layers of 0–10 and 10–20 cm, respectively.** (A) The shrub layer at 0–10 cm soil layer. (B) The shrub layer at 10–20 cm soil layer. (C) The herb layer at 0–10 cm soil layer. (D) The herb layer at 10–20 cm soil layer. The numerical serial number represents different kinds of plants. Refer to Tables 2 and 3 for the specific meaning. The figures show only the top ten plants in terms of relative abundance.

and *Urena lobate* (Linn.) was narrow. *Debregeasia orientalis*, a neutral plant, appeared in treatment C. A shade-loving plant, *L. pungens*, became more common in treatment C than in the other treatments. Niche breadth is a measure of the utilization of environmental resources by a population and can express the status and role of the population in the community. In this study, the niche breadth indicated that species and plant characteristics were more abundant under C treatment. Generally, after crop tree management on the plantation, due to the removal of large plants, larger canopy gaps exist, and the sufficient light is more conducive to the growth of light-loving plants (*Haughian & Frego, 2016*), both in space and time, and this may induce changes in abiotic factors (light, temperature and water) and biotic factors (bacteria and fungi) (*Lin, Zheng & Zheng, 2020*). However, our finding is contrary to those of other studies (*Schulze et al., 2016*; *Zhou et al., 2017*), perhaps because the density of the crop trees in the C treatment was the highest. Although the light intensity in the forest increased to a certain extent under treatment C, compared with the other treatments, the forest was dark and moist, resulting in a transitional environment for the plant pattern. If the operating life of the crop tree increased, it would change this pattern, and the whole woodland ecosystem would tend to be stable. The heterogeneity of light conditions and canopy structure, the maintenance of tree species richness and the

**Table 6  The eigenvalues and percent of variance explained.**

| Index | Principal component 1 | Principal component 2 | Principal component 3 |
|---|---|---|---|
| Eigenvalue | 9.029 | 2.973 | 1.448 |
| Explained/% | 60.194 | 19.822 | 9.654 |
| Cumulative variance explained/% | 60.194 | 80.016 | 89.67 |

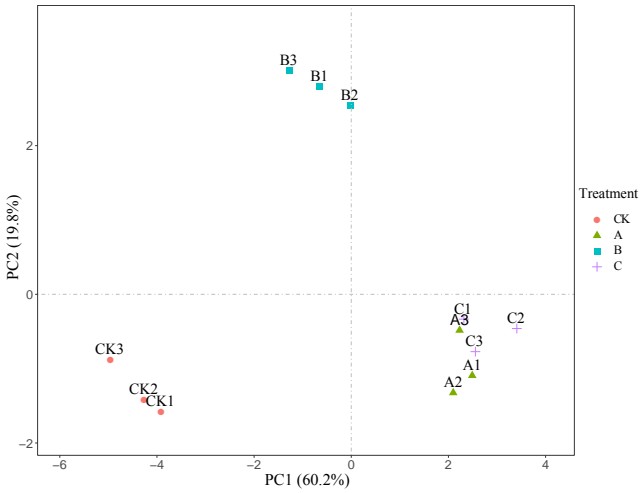

**Figure 6  Principal component analyses of crop tree management of *P. massoniana* plantation.** (A) Crop tree densities of 100 N/ha, (B) crop tree densities of 150 N/ha, (C) crop tree densities of 200 N/ha.

**Table 7  Comprehensive appraisal value and ranking in the early stage of crop tree management.** A (crop tree densities of 100 N/ha), B (crop tree densities of 150 N/ha), C (crop tree densities of 200 N/ha).

| Treatments | Principal component | | | Comprehensive evaluation | Rank |
|---|---|---|---|---|---|
| | F1 | F2 | F3 | F | |
| CK | −1.458 | −0.75 | 0.112 | −1.133 | 4 |
| A | 0.756 | −0.561 | −1.027 | 0.273 | 2 |
| B | −0.217 | 1.613 | −0.202 | 0.189 | 3 |
| C | 0.919 | −0.303 | 1.117 | 0.992 | 1 |

continuity of forest are the key factors affecting the diversity of forest herbaceous plants (*Marialigeti et al., 2016*), so the increase in the herb layer plant diversity index was small in treatment C.

Our results confirmed that crop tree management would increase the undergrowth plant diversity, significantly increase the nutrient elements in the topsoil, and optimize the forest environment, which is consistent with the results of most studies (*Lindgren & Sullivan, 2013*; *Zhou et al., 2016*). In the initial stage of crop tree management, human disturbance has a great influence. Through the transportation and migration of fallen trees in the forest, the soil surface layer is seriously affected, and the structure and function of

soil aggregates are changed (*Du et al., 2020*; *Teramage et al., 2020*). At the same time, the rapid decrease in the litter layer leads to a decrease in soil water storage and absorption and affects the mechanism of the soil cycle (*Capowiez et al., 2021*), so the soil bulk density of topsoil decreases significantly. The significant increase in soil moisture may be due to the decrease in water evaporation after tree thinning and the increase in precipitation in the canopy gap (*Bassett, Landis & Brudvig, 2020*). In our study, we found that the content of soil available potassium decreased significantly in the initial stage of crop tree management. This may have been due to the increase in light intensity after thinning around the crop tree increasing the demand for available potassium for plant growth, which would decrease the available potassium in the soil accordingly (*Wang et al., 2013*). In addition, available potassium often exists in the soil in the form of potassium ions, and ion exchange easily occurs, which also reduces the content of available potassium. Previous studies have shown that the initial stage of crop tree management does not have a significant impact on deep soil nutrients, and the results may be related to the operating life of crop trees.

## Interaction between plants and soil

After crop tree management, the diversity of herb layer increased significantly, probably because the herb layer plants are more sensitive than the shrub layer, most of them were annual plants, and the seeds are easy to reproduce (*Graf et al., 2019*; Durak & Durak, 2021). Plant diversity is an important factor that affects the soil food chain and soil functions (*Li et al., 2020b*). The content of nitrogen and phosphorus in soil is generally the limiting factor for plant growth (*Lucasborja & Delgadobaquerizo, 2019*), which is well confirmed in this study. On the one hand, available phosphorus has a very significant positive correlation with the diversity of the shrub and herb layers. The respiration of fine roots and the absorption of nutrients vary among plant species, which promotes changes in the soil organism community and strengthens the decomposition of litter. Large amounts of phosphorus enter the soil (*Yang, Maron & Callaway, 2015*). Hydrolysable nitrogen was only significantly positively correlated with the Pielou index of the herb layer. *Pinus massoniana* has a developed root system, the shrub and herb layers have high species richness, and the intertwining of roots leads to an increase in soil microbial activity and quantity. *Pinus massoniana* fine root biomass increased significantly in the early stage of crop tree management, which strengthened the soil nitrogen cycle (*Li et al., 2020a*). On the other hand, as the main environmental factors, the contents of soil hydrolysable nitrogen, available phosphorus and available potassium significantly affected the distribution of dominant species in the shrub and herb layers.

In the initial stage of crop tree management, the ecological environment of *Pinus massoniana* plantation was significantly improved, the number of different plant types was higher, and the natural plant regeneration was promoted. The C treatment had the most obvious effect on the increase in topsoil nutrients in *Pinus massoniana* plantation, and the strong effect of the plant seed bank in the forest was the most obvious. In this study, we observed that treatment C was significantly more effective than treatments A and B in the improvement of plant community composition and soil fertility. The forest ecosystem tended to develop towards becoming closer to a natural state, which further promoted seed

germination and reproduction in various plants. The reason for this result is that, compared with treatments A and B, treatment C retained more crop trees and required the removal of more interfering trees affecting crop tree growth; therefore, treatment C increased the resources and space available for plant habitat more than the other treatments. Generally, ecosystems are constrained by limited habitat resources, and plants can accelerate their growth and promote regeneration in more favourable habitats. Some studies have shown that cutting can promote plant growth and regeneration because of the increase in light availability after cutting (*Yang et al., 2018*). In the initial stage of crop tree management, the reduction in canopy density generates the most obvious changes in the forest environment in artificial forests, including more available light, increased precipitation and relatively increased growth space (*Chen et al., 2019*).

## CONCLUSION

The results showed that crop tree management in *Pinus massoniana* plantation with three different crop tree densities significantly increased the plant diversity of the forest understory and improved the physicochemical properties of the topsoil. The comprehensive evaluation was 200 N/ha >100 N/ha >150 N/ha >CK. A crop tree density of 200 N/ha was the most appropriate density for the initial stage of plantation crop tree management, which not only increased the undergrowth plant diversity, but also promoted the interaction between vegetation and soil. The crop tree density of 100 N/ha and 150 N/ha were not as effective as 200 N/ha, probably because of its small number of reserved trees, canopy gaps and light intensity that could not make plants grow stably in a short time. Our research provided a stronger theoretical and scientific basis for the reconstruction and management of *Pinus massoniana* plantations. To achieve the sustainable development of *Pinus massoniana* plantations and the accurate improvement of forest quality, it is necessary to continue this study as a long-term investigation.

**Abbreviations**

| | |
|---|---|
| **SOM** | Soil Organic matter |
| **AP** | Available potassium |
| **AN** | Hydrolysable nitrogen |
| **AK** | Available potassium |
| **SM** | Soil moisture |
| **BD** | Bulk density |

## ACKNOWLEDGEMENTS

We thank Tianchi Forest Farm in Huaying city for various forms of assistance, and we sincerely thank the forestry workers and instructors who worked hard for this experiment.

### Funding

This study was funded by a Pillar Project of the ''13th'' Five-Year Plan for China (grant number 2017YFD060030205), German Government Loans for Sichuan Forestry Sustainable Management (grant number G1403083) and the Study on Species Diversity of large diameter Timber Forest of Pinus massoniana supported by Tianfu Ten Thousand talents Plan of Sichuan Province (1922999002). The funders had no role in study design, data collection and analysis, decision to publish, or preparation of the manuscript.

### Grant Disclosures

The following grant information was disclosed by the authors:
A Pillar Project of the ''13th'' Five-Year Plan for China: 2017YFD060030205.
German Government Loans for Sichuan Forestry Sustainable Management: G1403083.
Tianfu Ten Thousand talents Plan of Sichuan Province: 1922999002.

### Competing Interests

The authors declare there are no competing interests.

### Author Contributions

- Qian Lyu conceived and designed the experiments, performed the experiments, analyzed the data, prepared figures and/or tables, authored or reviewed drafts of the paper, and approved the final draft.
- Yi Shen conceived and designed the experiments, performed the experiments, prepared figures and/or tables, authored or reviewed drafts of the paper, and approved the final draft.
- Xianwei Li and Gang Chen conceived and designed the experiments, prepared figures and/or tables, authored or reviewed drafts of the paper, and approved the final draft.
- Dehui Li and Chuan Fan conceived and designed the experiments, authored or reviewed drafts of the paper, and approved the final draft.

### Data Availability

The raw measurements are available in the Supplemental File.

### Supplemental Information

Supplemental information for this article can be found online at http://dx.doi.org/10.7717/peerj.11852#supplemental-information.

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
