# Peer review of "Early effects of crop tree management on undergrowth plant diversity and soil physicochemical properties in a Pinus massoniana plantation"

_PeerJ, doi:10.7717/peerj.11852_

## Round 0.1 · original submission · Major Revisions

Dear authors

Revise your paper while replying to each comment of both reviewers. Also, try to improve the English, and add some updated literature for the last five years.

Reviewer 1 ·

Basic reporting

This manuscripts describes the current status of Pinus massoniana plantations and to provide future direction to achieve sustainable forest management. Author studied a 36-year-old P. massoniana plantation in Huaying city of Sichuan Province. They investigated the changes in undergrowth plant diversity and soil physicochemical properties under three different crop tree densities (100, 150, and 200 N/ha) and concluded that crop tree management significantly increased the undergrowth plant diversity and that the shrub layer diversity was more delicate than the herb layer diversity to the treatments. Further, the contents of available phosphorus, organic matter and hydrolysable nitrogen in the topsoil increased significantly after crop tree management, while the content of available potassium decreased. This study revealed that treatments, 200 N/ha had the best effect on improving the undergrowth environment of the P. massoniana plantation in the initial stage of crop tree management as compared with the other different treatments as used in this study.
I would like to express appreciation to the authors for this study, which aims providing the suggestion for the improvement of Forest cop trees and ultimately to maintain the ecosystem and livelihood. I found the manuscript very interesting to read, but also came across few issues that will deserve some careful attention.

Experimental design

1. The article need to be revised fully, precisely focusing on the results presentation, discussion and conclusion.
2. Abstract should be revised carefully and adopt the scientific style …..
3. In title, add the authority with botanical name and as well in first place in MS.
4. Line 16-17…This statement is ambiguous….please revise it…………
5. Line 51….Please add here reference to support your statement……….
6. Line 55….revise these sentences….
7. Similarly…..as mentioned above few lines….there are many sentences which need to be revised….
8. The article should be revised logically to improve the results and discussion section particularly, conclusion is not good, should be revised.
9. This study revealed that treatments, 200 N/ha had the best effect on improving the undergrowth environment of the P. massoniana plantation in the initial stage of crop tree management as compared with the other different treatments as used in this study. This statement also indicate that the more amount of treatment is beneficial for crop growth, so would be suggested if author has used other different treatment should include, so clearly can draw the conclusion regarding treatment amount.
10. Overall, I recommend the article should be published after major revision of the papers results and English improvement.

Validity of the findings

1. The article need to be revised fully, precisely focusing on the results presentation, discussion and conclusion.
2. Abstract should be revised carefully and adopt the scientific style …..
3. In title, add the authority with botanical name and as well in first place in MS.
4. Line 16-17…This statement is ambiguous….please revise it…………
5. Line 51….Please add here reference to support your statement……….
6. Line 55….revise these sentences….
7. Similarly…..as mentioned above few lines….there are many sentences which need to be revised….
8. The article should be revised logically to improve the results and discussion section particularly, conclusion is not good, should be revised.
9. This study revealed that treatments, 200 N/ha had the best effect on improving the undergrowth environment of the P. massoniana plantation in the initial stage of crop tree management as compared with the other different treatments as used in this study. This statement also indicate that the more amount of treatment is beneficial for crop growth, so would be suggested if author has used other different treatment should include, so clearly can draw the conclusion regarding treatment amount.
10. Overall, I recommend the article should be published after major revision of the papers results and English improvement.

Additional comments

I would like to express appreciation to the authors for this study, which aims providing the suggestion for the improvement of Forest cop trees and ultimately to maintain the ecosystem and livelihood. I found the manuscript very interesting to read, but also came across few issues that will deserve some careful attention.

Reviewer 2 ·

Basic reporting

This manuscript "Early effects of crop tree management on undergrowth plant diversity and soil physicochemical properties in a Pinus massoniana plantation" is fairly well written and worth publishing in PeerJ but it needs revisions before it is published. The comments and suggestions are annotated in the manuscript. Authors are advised to get a native English speaker to correct English before re-submit.

Experimental design

Generally experimental design is fine.

Validity of the findings

The findings are valid and interesting.

Additional comments

This manuscript "Early effects of crop tree management on undergrowth plant diversity and soil physicochemical properties in a Pinus massoniana plantation" is fairly well written and worth publishing in PeerJ but it needs revisions before it is published. The comments and suggestions are annotated in the manuscript. Authors should get a native English speaker to correct English before re-submit.

Annotated reviews are not available for download in order to protect the identity of reviewers who chose to remain anonymous.

---

## Round 0.2 · accepted · Accept

The manuscript is improved tremendously and accepted for publication. One thing little confusing for me is what the authors mean by crop trees (explain). I understand trees are individual big plants, but a crop is a community of plants in field condition. Need clarification, thanks.


Reviewer 1 ·

Basic reporting

The authors completed extensive revision as per my comments and those of the other reviewer. I'm satisfied with the revision and encourage publication.

Experimental design

Its Ok.

Validity of the findings

The authors completed extensive revision as per my comments and those of the other reviewer. I'm satisfied with the revision and encourage publication.

Additional comments

The authors completed extensive revision as per my comments and those of the other reviewer. I'm satisfied with the revision and encourage publication.